# The Influence of a Cold Front and Topography on the Initiation and Maintenance of a Precipitation Convective System in North China: A Case Study

**DOI:** 10.3390/ijerph19159484

**Published:** 2022-08-02

**Authors:** Yan Li, Yu Wang, Xianyan Chen

**Affiliations:** 1Key Laboratory of Meteorological Disaster, Ministry of Education, Nanjing University of Information Science and Technology, Nanjing 210044, China; yanlee@nuist.edu.cn (Y.L.); 15189825252@163.com (Y.W.); 2National Climate Center, China Meteorological Administration, Beijing 100081, China

**Keywords:** convective rainfall, mesoscale convergence line, local terrain, Taihang Mountains

## Abstract

By using the convection-resolving weather research and forecasting simulation, a convective rainfall event over the middle portions of the eastern foothills of the Taihang Mountains in North China is investigated in this study. The influences of the cold front and complex topography on the initiation and maintenance of the convective system are analyzed. Results show two reasons why the convective clusters are initiated near noon on the hillsides at an elevation of 800 m. First, a local topographic convergence zone usually appears on the eastern slope of the Taihang Mountains near noon in May. Second, such a topographic convergence zone is enhanced by a cold front system and then triggers the convective clusters. Subsequently, the convective cells strengthen when moving downslope and weaken when moving eastward on the plain. When moving downslope, the atmospheric stratification is convectively unstable, and the mountain–plains solenoid (MPS) is strong near the foot of the mountain. The large amount of water vapor carried by the MPS-induced easterly wind is forced to ascend by topographic obstructions, and therefore the convective cells develop. As a result, heavy rainfall occurs on the hillsides with an elevation of 200–600 m. When the convective cells move eastward on the plain, the atmospheric stratification is stable, and the MPS is weak. Thus, convective activities weaken. Moreover, the results reveal that the mesoscale convergence line, slope gradient and slope aspect of the local terrain, local atmospheric instability, and the MPS play different roles in maintaining the convective system at elevations of 200–600 m along the eastern foothills of the Taihang Mountains.

## 1. Introduction

Under the rapid urbanization in China, disasters including heavy precipitation, floods, and urban waterlogging have become more serious, and disaster prevention and mitigation have become more and more difficult [1]. The annual average frequency of heavy rainfall in North China is lower than in South China, but heavy rainfall in North China is always accompanied by high intensity of short-term precipitation [2,3]. The Taihang and Yanshan Mountains, with altitudes of 800–2000 m, are located along the western and northern margins of North China, respectively [4]. The south–north-oriented Taihang Mountains can influence the low-level southeasterly and easterly jets, and the east–west-oriented Yanshan Mountains can affect the low-level southwesterly jet [5]. Therefore, heavy precipitation mainly occurs near the mountains close to the North China Plain [6,7].

The formation and development of heavy precipitation are found to be closely related to the interaction of various weather systems under favorable atmospheric conditions, and are affected by both the large-scale circulation systems and the thermal–dynamic effects of mesoscale systems. Therefore, the formation and development mechanisms of heavy precipitation are complex [5,8]. Numerous studies on heavy rainfall in North China concluded that topographic uplift, cold front system, mesoscale convergence line and vortex, dry intrusion, non-uniform humidity saturation, and “train effect” play essential roles in convection triggering [9,10,11,12,13,14,15]. Dynamic processes, such as conditional symmetric instability, inertial gravity instability, and the interaction between convective storms and low-level jets, have a great impact on the intensification and maintenance of convection [14,16,17]. Cold fronts are one of the important influence systems for the heavy rainfall in North China. It has the typical baroclinic atmospheric structure, with a stronger horizontal temperature or humidity gradient. Mesoscale convergence line, dry line, and the secondary circulation of cold fronts are also crucial for triggering convection in North China [18,19,20].

There are two types of topographic precipitation, namely, the windward slope rainfall caused by the topographic uplifting effect [21] and the rainfall caused by the mountain–plains solenoid (MPS) that is induced by the uneven solar heating [22]. During the daytime, the air temperature on the mountain top increases faster than that of the surrounding air, which is conducive to the convergence near the mountain top and the initiation of convection. In the afternoon, the ascending branch of the MPS is located near the mountains and promotes the precipitation in the mountain areas, while in the evening, the ascending branch is located in plains or basins and facilitates the precipitation in plains or basins [4,23,24]. The topography of North China is complex, and the topographic forcing is closely related to the heavy rainfall in North China [10,25]. Past studies have shown that lifting of warm and wet low-level flow forced by mountains is beneficial to the heavy precipitation on the eastern slope of mountains near the plain [6,11]. Hua et al. (2020) [25] analyzed the influence of multi-scale topography on the formation and maintenance of precipitation convection systems in North China. The results show that convection is triggered by the small-scale MPS between mountains and basins, and its maintenance and development depend on the large-scale MPS between plateaus and plains. In addition, the topography of the Taihang Mountains also has an important impact on water vapor transport. Wang et al. (2014) [26] and Zhao et al. (2020) [27] suggested that the interaction between the topography of the Taihang Mountains and water vapor convergence enhances heavy precipitation processes in North China.

From 22–23 May 2017, a heavy rainfall event occurred along the eastern foothills of the Taihang Mountains in North China, with 12 h accumulated rainfall of 65 mm in Shijiazhuang, located in the middle portions of the eastern foothills of the Taihang Mountains. The cold front and the forcing of complex terrain are the main reasons for this convective rainfall. However, why does such convective rainfall mainly occur in the eastern foothills of the Taihang Mountains? What are the impacts of the Taihang Mountains and cold front on rainfall amount? To address these questions, we conduct some convection-resolving numerical simulations to analyze the mesoscale features of atmospheric circulation in this study. Furthermore, the influences of the cold front and topography on the initiation and maintenance of the convective system are investigated. These analyses contribute to understanding the development of precipitation convective systems in the eastern slope of the Taihang Mountains and forecasting the short-term heavy rainfall events.

The remainder of this paper is organized as follows. Section 2 describes the datasets and the setup of the numerical simulations used in this study. Section 3 provides an overview of this heavy rainfall event, including the environment conditions and the evolution of the convective system. The initiation and maintenance of the convective system are investigated in Section 4. A summary and conclusions are presented in Section 5.

## 2. Data and Model

### 2.1. Datasets

The datasets used in this study are listed in Table 1. The hourly precipitation data from 16 conventional automatic weather stations and 288 intensive automatic weather stations in Shijiazhuang within the range of 37.4° N–38.8° N, 113.5° E–115.5° E are selected to analyze the spatiotemporal distribution characteristics of precipitation. The 6 h FNL (Final Operational Global Analysis) data released by the National Center for Environmental Prediction are used to analyze large-scale atmospheric conditions, with a horizontal resolution of 1° × 1° and 31 pressure levels from 1000 hPa to 1 hPa. The hourly and monthly average data from the EAR5 reanalysis dataset released by the European Center for Medium-Range Weather Forecasts are used to analyze the diurnal variations, with a horizontal resolution of 0.25° × 0.25° and 37 pressure levels from 1000 hPa to 1 hPa. In addition, the S-band Doppler radar (SA radar) data in Shijiazhuang have a resolution of 1 km.

### 2.2. Setup of the Numerical Model

The Weather Research and Forecasting (WRF) model is a fully compressible and non-hydrostatic grid model, widely used in the simulations of convective weather [28,29]. In this study, the WRF V4.1.1 model is adopted, and the 1° × 1° FNL data are used as the initial and lateral boundary conditions. The integration time is from 1400 LST (local solar time, hereinafter) on 21 May to 1400 LST on 23 May 2017, and the first 18 h are the spin-up time. The time interval of outputs is 1 h. The WRF model is configured with three two-way nested domains (Figure 1), and the grid center is set at 38.0° N, 114.0° E. The horizontal grid intervals are 27 km (D01; mesh size of 202 × 202), 9 km (D02; mesh size of 202 × 232), and 3 km (D03; mesh size of 199 × 199). The outermost domain covers most of China, the South China Sea, and part of the Pacific Ocean, while the innermost domain covers the central and southern parts of Hebei. There are 55 vertical levels in total and 9 levels below 1 km, and the lowest level is at about 10 m. The model top is set at 100 hPa. Model physics options, including the WSM5 microphysical scheme [30], the Grell–Devenyi ensemble for cumulus parameterization scheme [31], the YSU scheme for the planetary boundary layer [32], the five-layer thermal diffusion scheme for the slab soil model [33], the rapid radiative transfer model (RRTM) scheme for longwave radiation [34], and the Dudhia scheme for shortwave radiation [35], are used in all three domains.

## 3. Case Overview

### 3.1. Environment Conditions

From 0800 LST on 22 May to 0800 LST on 23 May 2017, there were six stations with 24 h accumulated rainfall exceeding 50 mm, and the 12 h accumulated rainfall exceeded 60 mm in some stations (Figure 2a). Figure 2b shows the temporal variation of rainfall observed at four intensive automatic weather stations in Shijiazhuang (B0877, B1229, B0882, and B1041). It can be seen that the rainstorm process is characterized by short duration, and the heavy rainfall mainly concentrated from 1200 LST to 1400 LST on 22 May. The rainfall peak occurred first at station B1041 in the south, then at stations B0882 and B1229 in the north, and finally at station B0877 in the far north.

The synoptic conditions at 0800 LST on 22 May 2017 are presented in Figure 3, when it was a few hours before convection initiation. It can be found that the main synoptic systems are the cold trough in the middle troposphere, the shear line in the lower troposphere, and the surface cold front. At 500 hPa (Figure 3a), the two ladder troughs over the Baikal Lake and Inner Mongolia led the cold air to move southward. The Northwest Pacific subtropical high was located southward, with the average ridge line at 15° N and the ridge point extending westward to 95° E. Subsequently, the Northwest Pacific subtropical high advanced northward stably, blocking the southward movement of the middle-latitude trough (figure not shown). The 850 hPa shear line between the continental high and Northwest Pacific subtropical high facilitated the initiation of convection (Figure 3b). The northwesterly wind from the northwestern edge of the continental high transported dry–cold air to the central–southern parts of Hebei Province. The southwesterly wind from the southeastern edge of the Northwest Pacific subtropical high provided the study region with warm–wet air. At this time, the surface cold front was oriented in a northeast–southwest direction, whose northern end was just located at the northwest edge of the convective system (Figure 3c).

### 3.2. Evolution of the Convective System

Figure 4 shows the evolution of the convective clusters from radar composite reflectivity on 22 May 2017. In this study, a convective cluster is defined as an aggregation of several close convective cores [6]. At 1142 LST, the convective cluster A formed and developed on the eastern slope of the Taihang Mountains at the elevation of about 600–800 m (Figure 4a). From 1142 LST to 1212 LST, cluster A, which was in the warm sector ahead of the cold front, moved slightly northward. Meanwhile, a new convective cluster B initiated on the Taihang Mountains at about 1000 m elevation to the west of cluster A (Figure 4b). After initiation, cluster A moved northward, while cluster B moved eastward. As a result, convective clusters A and B gradually approached each other. At 1236 LST, cluster B propagated to the elevation of about 800 m and produced heavy rainfall of 13.1 mm∙h^−1^ at station B0882 when its size and intensity both increased (Figure 4c and Figure 2b). One hour later, cluster A moved to B0877 in the north, leading to heavy rainfall with the intensity of 15.3 mm h^−1^. After that, cluster A continued to move northward and weakened while moving downhill, and cluster B moved eastward and southward following the cold front (Figure 4e,f).

## 4. Results

### 4.1. Model Verification

The radar echoes in North China at 0800 LST and 1400 LST on 22 May 2017 between observations and simulations suggest that there was a northeast–southwest-oriented rain belt caused by the cold front at 0800 LST in Jilin, Inner Mongolia, and Hebei Province. Compared with the observation, the area of simulated rain belt was slightly larger, but the distributions of convective cores were consistent, both located in the northern Hebei Province. At 1400 LST, the simulated rain belt moved southward to the central–southern Hebei Province, and the moving speed and direction were basically the same as those of observations. For the comparison of the 24 h accumulated precipitation between observations (Figure 2a) and simulations (Figure 5), it can be found that although the simulated area of heavy rainfall was larger, the distribution and intensity of the simulated rainfall were basically consistent with the observations. Overall, this simulation can generally reproduce the synoptic-scale characteristics of this precipitation process.

In order to investigate the simulation capability on the convective clusters, we further analyze the evolution of the composite reflectivity of the convective systems from 1130 LST to 1700 LST on 22 May 2017 (Figure 6). The simulated cluster A (i.e., convective cluster A’ in simulation) was initiated at 37.5° N at an elevation of 800 m on the eastern slope of the Taihang Mountains (Figure 6a), but was about 20 km to the east of the observed cluster A and the initiation time was about 10 min earlier than observation (Figure 4a). At 1330 LST, cluster A’ moved downslope to around 200 m elevation (Figure 6c). During this process, A’ moved northward and grew both in size and intensity, leading to heavy rainfall in the north, in agreement with observations, but A’ had a faster downslope speed and a more eastward spread.

The other initial convection (cluster B’ in simulation and cluster B in observation) appeared at about 800 m elevation to the southwest of A’ at 1230 LST (Figure 6c), which was to the south and east of cluster B and occurred later than observations (Figure 4b). In the following time, cluster B’ expanded and enhanced while moving downslope, which was consistent with the evolution of B in the observation. From 1400 LST to 1600 LST, the main part of A’ extended northward, and B’ enhanced and expanded after merging with the southern part of A’ (Figure 6f,j). After that, B’ continuously moved southward and weakened (Figure 6k,l). There are some spatiotemporal deviations between our simulations and observations, but these deviations commonly exist in numerical simulations [36]. Overall, our simulations can still reflect the initiation and development of the convective systems. Therefore, the simulation results are considered to be reliable in studying the development and evolution of the convective systems.

### 4.2. Convection Initiation

Convection initiation (CI) refers to a process in which air particles are uplifted above the level of free convection (LFC) after obtaining and maintaining positive buoyancy and finally form a deep convective cell [37]. Studies indicate that when the radar echo intensity is greater than 35 dBZ for the first time, it is determined that the CI has occurred [38,39,40]. Hence, the CI time was 1130 LST (A’) and 1230 LST (B’) on 22 May. Therefore, the period of 1030–1230 LST is selected to study the CI mechanisms.

Firstly, the CI occurred near noon, which may be associated with the local climate characteristics. Figure 7 illustrates the climatological 925 hPa diurnal variations of temperature anomaly and divergence anomaly (minus the daily average) in May from 1981 to 2010. It can be seen that there were great spatial differences in the near-surface temperature because of the different solar heating between mountains and the plain in the east. With the radiation increase during the daytime, the air over the elevated terrain was warmed more rapidly than over the plain. At 1200 LST, the south–north-oriented temperature contours over the eastern slope of the Taihang Mountains were distributed along the mountains, and the temperature gradient was the largest in the east–west direction (Figure 7c). The convective systems initiated exactly at noon at near 800 m elevation where both the east–west-oriented temperature gradient and the topographic slope were larger. This is because the increased temperature gradient intensified the local easterly airflow, which strengthened the convergence at the specific topographic structure under the dynamic uplifting of the local orography, forming the topographic convergence zone (Figure 7c). This climatic convergence zone is consistent with the frequent convective systems on the eastern slope of the Taihang Mountains [41,42,43].

Secondly, the CI occurred under favorable synoptic background. Figure 8 shows the wind field and divergence at 900 hPa on 22 May 2021. At 1130 LST, the strong horizontal temperature gradient enhanced the southeasterly wind to the south of cluster A’ and the easterly wind to the north of cluster A’ (Figure 8b). As a result, the intensity of the above topographic convergence zone was enhanced and A’ was triggered in it. At 1230 LST, with the gradual southward movement of the cold front (Figure 4c), the northeasterly wind to the north of A’ enhanced. A bow-shaped south–north-oriented convergence belt formed at the height about 400–800 m in the east of the Taihang Mountains. The two convergence cores were just corresponding to the development of A’ and the formation of B’. Thus, this CI was closely related to both the climatic terrain convergence zone and the cold front. It is also indicated that the area at the height of about 800 m in the east of the Taihang Mountains is favorable for CI.

### 4.3. Maintenance of the Convective System

#### 4.3.1. Atmospheric Conditions

As mentioned in the previous section, this precipitation process was produced by the joint effect of the 500 hPa westerly trough, 850 hPa shear line, and surface cold front. In this section, we focus on the main characteristics of large-scale circulations and mesoscale disturbances in the middle and lower troposphere. Figure 9 shows the simulated composite reflectivity, 850 hPa wind field, and cold front at the convection development stage. Figure 10 illustrates the perturbation wind field and divergence at 850 hPa.

In terms of the large-scale atmospheric circulations, the surface cold front moved southward at about 400 km∙h^−1^. The convective cluster A’ was located at 850 hPa near 400 km in front of the cold front at 1300 LST (Figure 9a). Subsequently, A’ and B’ moved eastward and enhanced. At 1400 LST, the section of cold front in the north of the convective clusters moved to 37.5° N and merged with the convective clusters (Figure 9b). Under the dynamic uplift of the cold front, the intensity and size of the convective system increased. As they moved southward at about 300 km∙h^−1^, cluster A’ merged into cluster B’. As a result, the merged system kept intensifying and reached its maximum size at 1500 LST. At 1600 LST, the section cold front in the eastern slope of the Taihang Mountains moved to near 37° N, and the main part of the convective cluster was near 37.2° N (Figure 9d). The convective cluster decreased gradually when it was located in the cold section after the cold front.

In terms of mesoscale systems, it can be found that there was a significant mesoscale convergence line at 850 hPa. At 1300 LST, a weak mesoscale cyclonic anomaly (denoted by “C” on Figure 10) appeared in the cold front area on the eastern slope of the Taihang Mountains, with the 850 hPa center located near 37.9° N, 113.6° E (Figure 10a). At this moment, convective clusters A’ and B’ were located at the topographic convergence center where southerly wind prevailed. An hour later, the wind around and north of A’ and B’ deflected counterclockwise, resulting in the strengthening of convergence near the convective systems. Therefore, A’ and B’ developed continuously (Figure 10b). Subsequently, the convective clusters were quasi-stationary, while the cyclonic anomaly accompanied by a cold front moved southward. Hence, the northeasterly anomaly in the north of convective clusters A’ and B’ became stronger, leading to the enhancement of convergence to the south. As a result, an east–west-oriented convergence line, which connected convective cluster B’ with the cyclonic anomaly, formed at 1500 LST (Figure 10c,d). At this time, the convective clusters developed most vigorously. Therefore, the low-level convergence line generated by the interaction between the topographic convergence and the cold front plays a key role in the rapid development of the convective clusters.

The conditionally unstable stratification in the middle and lower troposphere is an essential condition for the development of a convective system [44,45]. In this study, the skew-T log-P diagrams from simulations at station B0877 at 0800 LST and 1000 LST before rainfall are selected to analyze the local atmospheric stratification (Figure 11). The results show that the low-level atmosphere below 600 hPa was relatively humid, while a relatively dry layer presented above 600 hPa, especially at 500 hPa. There was a weak warm advection from 850 hPa to 600 hPa, a weak cold advection from 600 hPa to 400 hPa, and a weak inversion layer from 700 hPa to 600 hPa. The convectively unstable air columns in the middle and lower troposphere were greatly conducive to convective weather. In addition, the convective available potential energy (CAPE) at 0800 LST and 1000 LST was only 5 J∙kg^−1^ and 108 J∙kg^−1^, respectively. The southwesterly wind at low-level was also weak, about 4–6 m∙s^−1^, which means that the water vapor transport in the atmosphere is weak. Overall, this convective weather occurred in the unstable atmospheric environment with dry–cold air in the middle levels and warm–moist air in the low levels, but the CAPE was small. This situation is different from typical deep moist convection in southeastern China [28,29,46], but it is similar to the research results in North China from Hua et al. (2020) [25].

#### 4.3.2. Impact of the Complex Topography

Precipitation in central North China is significantly related to the topography [6,10,25]. Figure 12 and Figure 13 present the zonal-vertical cross-sections of the radar echo, potential temperature, equivalent potential temperature, and water vapor flux at centers of the convective cells (black lines in Figure 7) at the convection development stage. Figure 12 suggests that the cloud top height of convective clouds was below 400 hPa at the convection development stage, indicating that it was a shallow convective system. With the southward movement of the cold front, the convective cells moved downslope from the west to the east at the speed of 200 km∙h^−1^ and gradually weakened after moving to 114.8° E. The convective cells intensified downslope in the eastern slope of the Taihang Mountains and weakened when moving eastward on the plain. This characteristic is related to the local atmospheric stratification. From 1300 LST to 1400 LST, the local atmosphere had a convective instability below 600 hPa (Figure 13a,b), which is consistent with the analysis results in Figure 12. Thereafter, the instability gradually weakened. At 1500 LST, only the atmosphere on the hillside displayed convective instability, and the atmospheric stratification changed to be stable over the plain, which was one of the main reasons for the gradual weakening of convective cells when moving eastward on the plain (Figure 13c).

The characteristic that the convective cells enhanced when moving downslope and weakened when moving eastward on the plain area was also closely associated with the circulation variations in the eastern Taihang Mountains and the North China Plain. From 1300 LST to 1600 LST, the low-level easterly wind over the plain on the east of the Taihang Mountains gradually formed and strengthened, and the formation of the easterlies was related to the MPS caused by the difference of solar heating between the Taihang Mountains and the North China Plain. Figure 14 shows the cross-sections of vertical velocity and the anomalous wind and vertical velocity with daily mean values removed. As can be seen, the vertical velocity anomaly (shaded) had a magnitude comparable to the observed vertical velocity (contours), indicating that the vertical motion was mainly affected by the upward branch of the MPS [23]. The MPS caused an obvious circulation structure with anomalous easterly inflow, upward motion, and westerly outflow near the foothills of the mountain (Figure 14), and the upward branch of the circulation exactly corresponded to the development of convective cells (Figure 12). At 1400 LST, the easterly wind in the lower troposphere gradually formed, with a speed of about 0–2 m∙s^−1^ (Figure 13b). At 1500 LST, the easterly wind rapidly intensified to 10–16 m∙s^−1^, the horizontal range expanded to 115.4° E, and the vertical range extended to above 900 hPa. Under the effect of easterly anomalies, the water vapor transport increased significantly (Figure 13c and Figure 14c). Meanwhile, the upward motion strengthened markedly with the maximized at 800–600 hPa, indicating the development of local MPS. As a result, the convective cells near the foothill developed rapidly (Figure 12c). At 1600 LST, the easterly wind extended to about 115.8° E and began to weaken, leading to the decrease of water vapor transport (Figure 13d and Figure 14d) and the weakening of convection (Figure 12d). Thus, the anomalous easterly wind significantly influenced the convection development over the eastern slope of the Taihang Mountains. First, the combined effect of the easterly wind over the plain and the terrain of the eastern slopes of the Taihang Mountains enhanced the dynamic uplifting for the convection development near the foothills. Second, the anomalous easterly wind strengthened the low-level water vapor transport. The results are consistent with the previous study [6].

In addition, during the eastward movement of convective cells, there were new convective cells continuously initiated on the eastern slope of the Mountains (Figure 12). These initiations were associated with the downdraft and cold pool in front of convective cells. When the downdraft reached the surface, it was blocked by the cold pool and easterly inflow, and was forced to ascend during the westward movement (Figure 13). The dynamic uplifting effect of the topography and the local unstable stratification strengthened the local upward motion. Therefore, new convective cells were constantly formed on the hillside, and then moved eastward and merged with the convective clusters (Figure 6d–j).

## 5. Conclusions

On 22 May 2017, under the combined effect of the upper-level cold trough at 500 hPa, the shear line at 850 hPa, the cold front near the surface, and the 850 hPa convergence line, heavy convective precipitation occurred at the height of about 200–800 m on the foothills of the Taihang Mountains in North China. In this study, the high-resolution WRF model was used to carry out numerical simulations to analyze the combined effect of topography and cold front on the initiation and development of mesoscale convective systems. The conceptual model is summarized in Figure 15.

From morning to noon on 22 May, because of the difference of land surface characteristics between the mountains and the plain in the east, the near-surface climatic isotherms over the foothills of the Taihang Mountains were distributed along the mountains. The east–west-oriented horizontal temperature gradient gradually increased and reached the maximum around noon, then the local easterly airflow was triggered. A topographic convergence zone was formed when the easterly wind was blocked by the local terrain. On the other hand, from 1130 LST to 1230 LST on 22 May, the approach of the cold front enhanced the northeasterly wind to the east and north of the convective clusters, leading to further enhancement of the topographic convergence zone. As a result, the two convergence cores were just corresponding to the development and formation of the two initial convective clusters (Figure 15a). The convective systems initiated exactly at noon at about 800 m, indicating that the initiation of convective systems was closely related to local topography.

As a result, the intensity of the above topographic convergence zone was enhanced, and A’ was triggered in it. At 1230 LST, with the gradual southward movement of the cold front (Figure 4c), the northeasterly wind to the north of A’ enhanced. A bow-shaped south–north-oriented convergence belt formed at the height about 400–800 m in the east of the Taihang Mountains. The two convergence cores were just corresponding to the development of A’ and the formation of B’. Thus, this CI was closely related to both the climatic terrain convergence zone and the cold front. It is also indicated that the area at the height of about 800 m in the east of the Taihang Mountains is favorable for CI.

Under the atmospheric conditions of low CAPE and unstable stratification, the mesoscale convergence line, which was generated by the interaction of the lower-level topographic convergence and cold front, played a key role in the rapid development of convection clusters. When the convective cells moved downslope in the eastern slope of the Taihang Mountains, they intensified and expanded. Until moving eastward on the plain area, the convective cells weakened. The variation characteristics of convection intensity were closely related to the local atmospheric stratification and MPS (Figure 15b). Firstly, although the lower-level atmosphere was convectively unstable on the hillside, the lower-level atmosphere on the eastern plain was gradually transformed to stable at a later stage. Secondly, the easterly anomalies and the upward branch of MPS developed at 1500 LST. The MPS tended to weaken at 1600 LST, and the easterly anomalies and the upward motion also weakened. Therefore, when the convective cells moved downslope, the MPS-induced easterly anomalies and the local unstable convective environment enhanced the dynamic uplift effect near the foothills of the Taihang Mountains and the water vapor transport on the plain, thereby resulting in further development of convective cells. When the convective cells moved to the plain, the local atmospheric stratification was stable, and the MPS system weakened. Thus, the convection gradually weakened.

Certainly, the above conclusions still need to be further examined. For example, what are the effects of the different slope gradients and slope aspects in the middle of the Taihang Mountains on the initiation and development of convection, and do different mountain scales have different effects of MPS on convection development? Further sensitivity experiments should be designed in the future. Moreover, it is necessary to further investigate cloud-permitting models in the future in order to obtain further insight into the exact initiation and maintenance mechanisms of convection under the effect of complex topography, e.g., hillsides with altitudes of 200–600 m.

## Figures and Tables

**Figure 1 ijerph-19-09484-f001:**
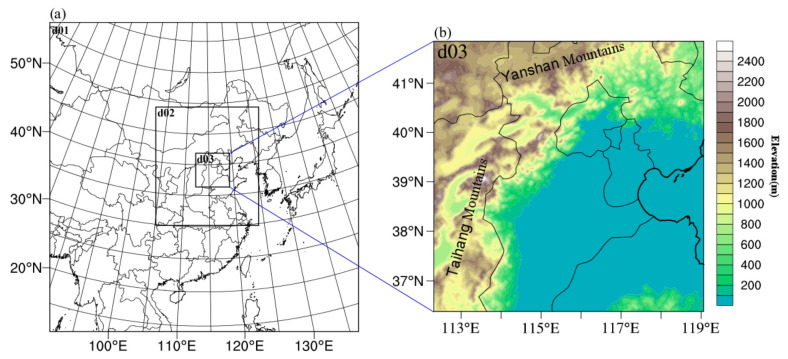
(**a**) The nested domains used in the simulation and (**b**) the surface elevation of d03. The grid intervals of d01, d02, and d03 are 27, 9, and 3 km, respectively. The shaded area shows the elevation (m) from the U.S. Geological Survey 30 s global dataset (GTOPO30).

**Figure 2 ijerph-19-09484-f002:**
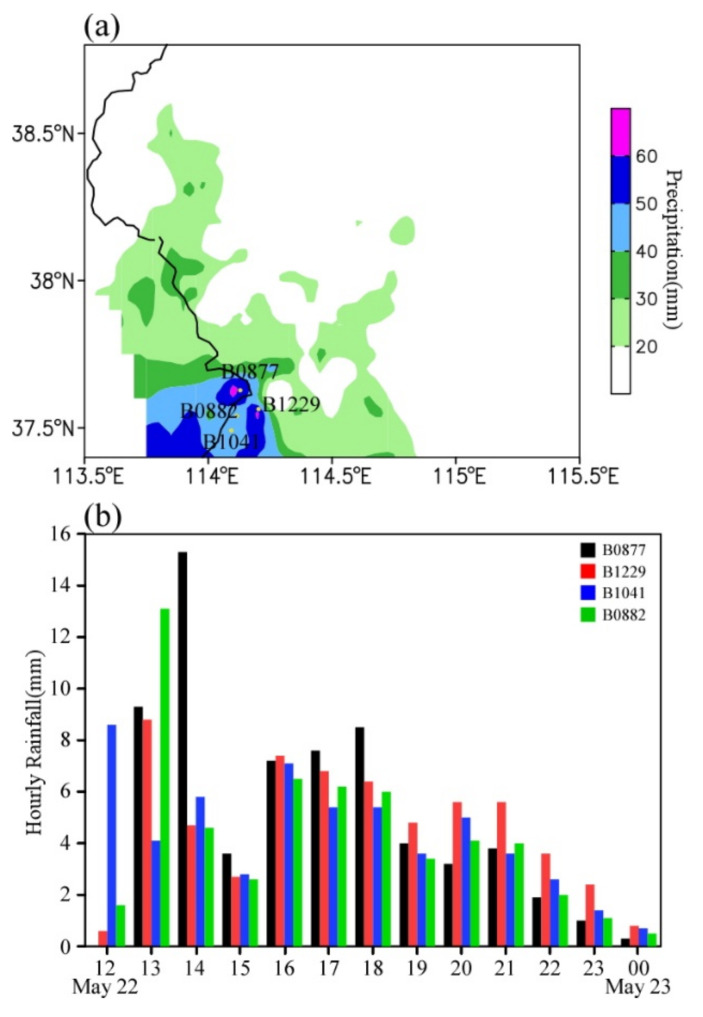
(**a**) The spatial distribution of accumulated rainfall from 0800 LST on 22 May to 0800 LST on 23 May 2017 (mm), and (**b**) the time series of rain gauge observations (mm h^−1^) at B0877 (black), B1229 (red), B1041 (blue), and B0882 (green) during the period from 1200 LST on 22 May to 0000 LST on 23 May 2017. The points in (**a**) represent the locations of four automatic weather stations in Shijiazhuang (B0877, B1229, B0882, and B1041), and the black line denotes the province boundary.

**Figure 3 ijerph-19-09484-f003:**
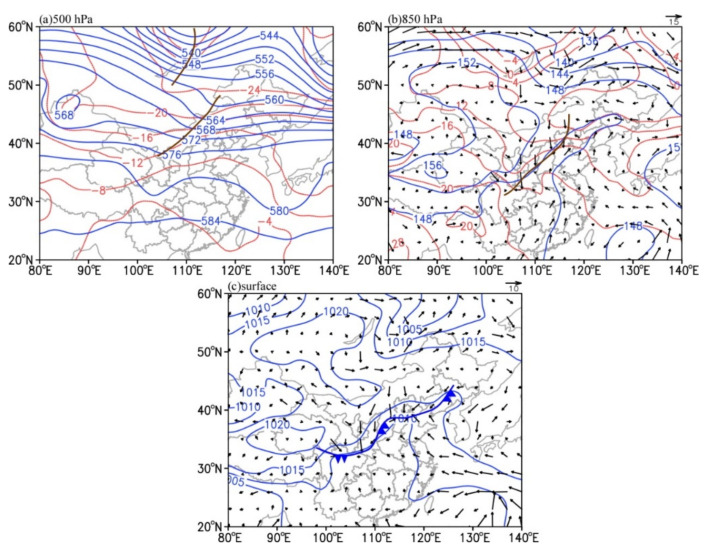
Synoptic conditions at 0800 LST on 22 May 2017: (**a**) Temperature (red contour, °C) and geopotential height (blue contour, dagpm) at 500 hPa; (**b**) temperature (red contour, °C), geopotential height (blue contour, dagpm), and wind field (vector, unit: m s^−1^) at 850 hPa; (**c**) sea level pressure (blue contour, hPa) and 10 m wind field (vector, unit: m s^−1^). The brown lines in (**a**,**b**) indicate the 500 hPa trough and the 850 hPa shear, respectively. The blue solid line with arrows in (**c**) denotes the position of the cold front at the surface.

**Figure 4 ijerph-19-09484-f004:**
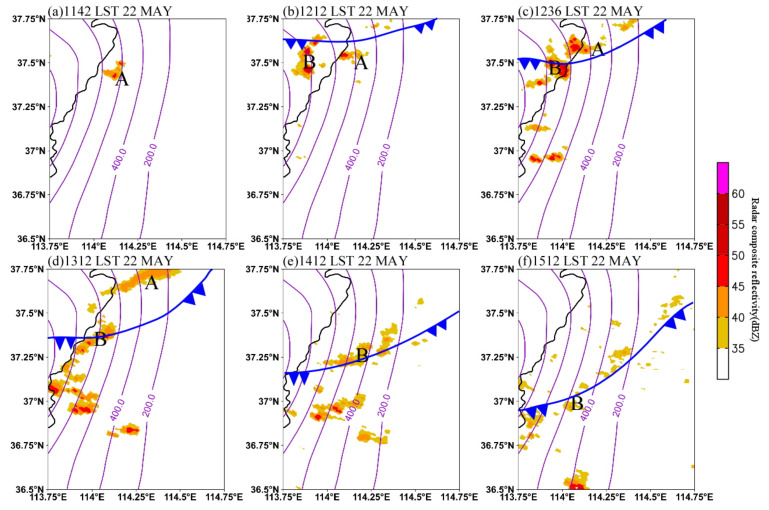
Radar composite reflectivity (dBZ) at (**a**) 1142 LST, (**b**) 1212 LST, (**c**) 1236 LST, (**d**) 1312 LST, (**e**) 1412 LST, and (**f**) 1512 LST on 22 May 2017. Letters “A” and “B” indicate the isolated convective systems, and blue solid line with arrows denotes the position of a cold front at the surface. The topography is contoured from 200 m to 1200 m at an interval of 200 m.

**Figure 5 ijerph-19-09484-f005:**
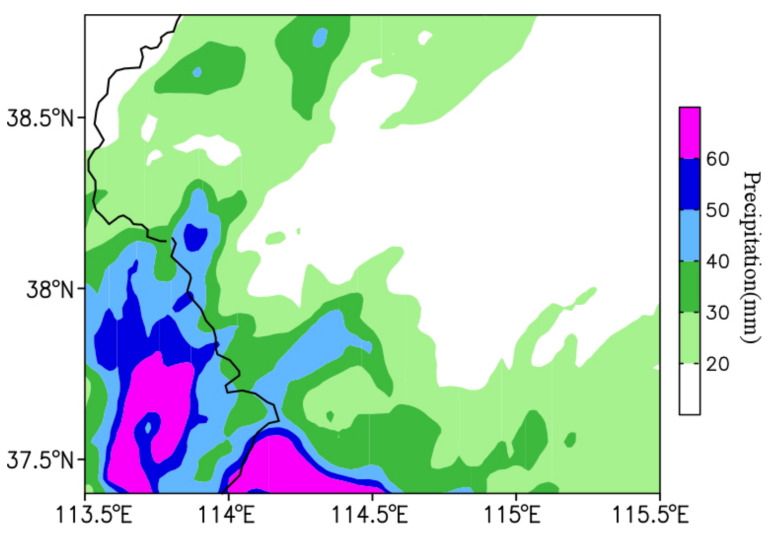
The simulated cumulative rainfall (mm) from 0800 LST on 22 May to 0800 LST on 23 May 2017.

**Figure 6 ijerph-19-09484-f006:**
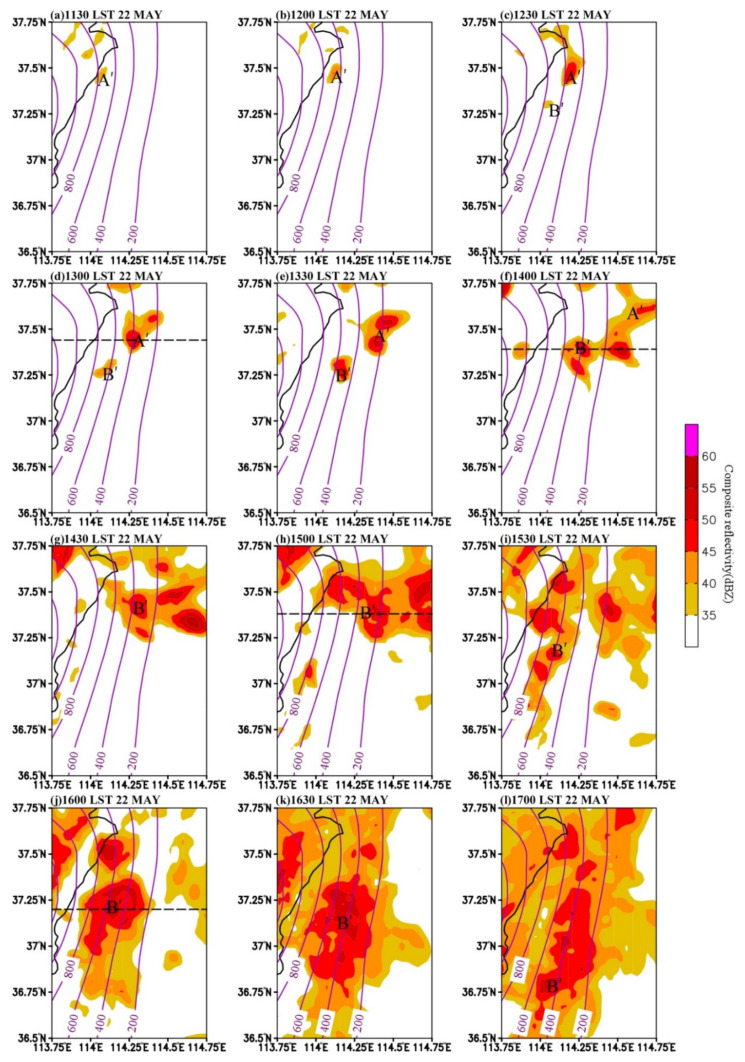
Simulated composite reflectivity (dBZ) at (**a**) 1130 LST (**b**) 1200 LST, (**c**) 1230 LST, (**d**) 1300 LST, (**e**) 1330 LST, (**f**) 1400 LST, (**g**) 1430 LST, (**h**) 1500 LST, (**i**) 1530 LST, (**j**) 1600 LST, (**k**) 1630 LST, and (**l**) 1700 LST on 22 May 2017. The topography (unit: m) is contoured from 200 m to 1200 m at an interval of 200 m. The approximate location of the convective systems are indicated by the letters A’ and B’.

**Figure 7 ijerph-19-09484-f007:**
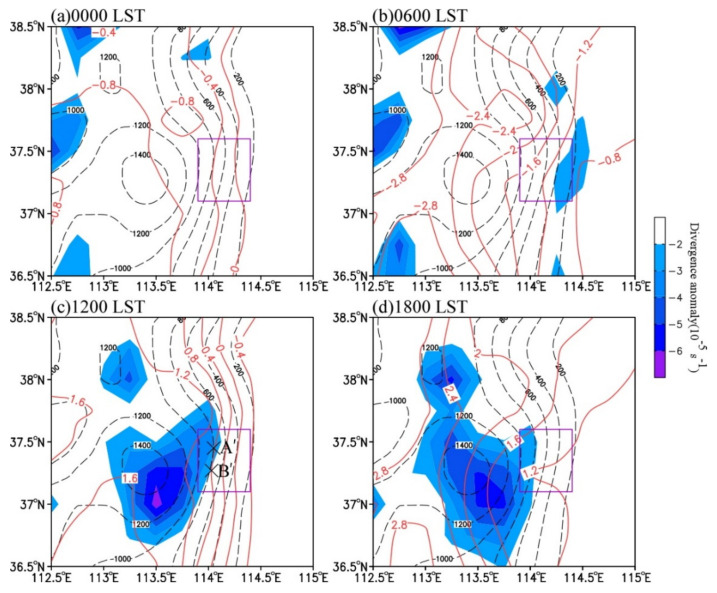
The climatological 925 hPa diurnal variations of temperature anomaly (with the daily mean removed; contour, °C) and divergence anomaly (with the daily mean removed; shaded, 10^−5^ s^−1^) at (**a**) 0000 LST, (**b**) 0600 LST, (**c**) 1200 LST, and (**d**) 1800 LST in May from 1981 to 2010. The purple box shows the main areas of convective initiation and development, and the symbol “×” in Figure 8c represents the initiation positions of A’ and B’.

**Figure 8 ijerph-19-09484-f008:**
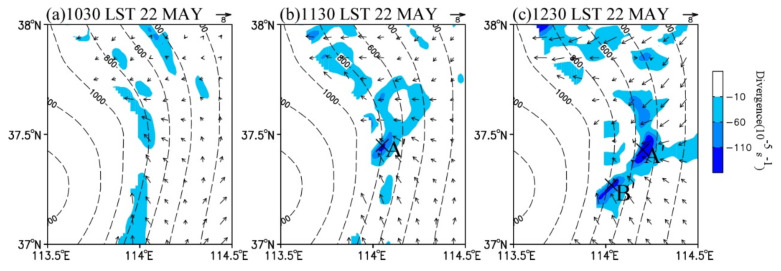
Simulated 900 hPa wind field (vector, m s^−1^) and divergence (shaded, 10^−5^ s^−1^) at (**a**) 1030 LST, (**b**) 1130 LST, and (**c**) 1230 LST on 22 May 2017. The symbol “×” indicates the position of A’ or B’.

**Figure 9 ijerph-19-09484-f009:**
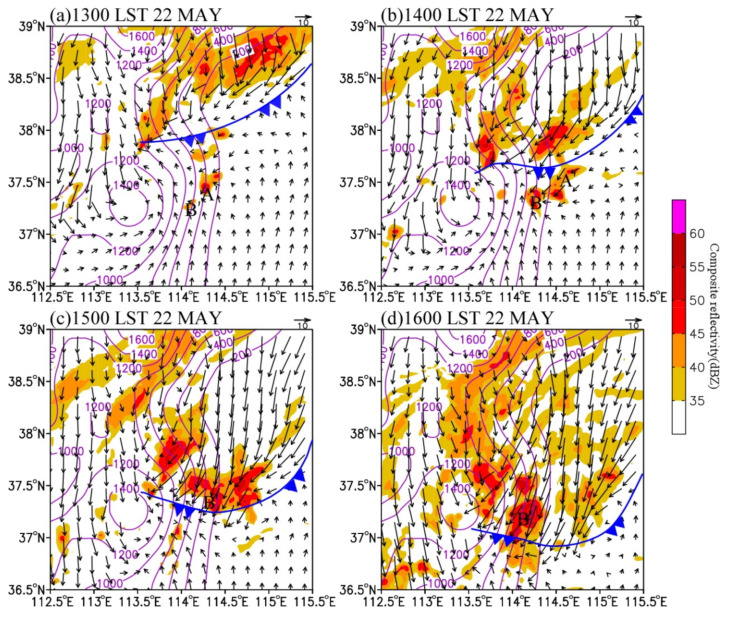
Simulated 850 hPa composite reflectivity (shaded, dBZ) and wind field (vector, m s^−1^) at (**a**) 1300 LST, (**b**) 1400 LST, (**c**) 1500 LST, and (**d**) 1600 LST on 22 May 2017. The surface cold fronts are shown in blue.

**Figure 10 ijerph-19-09484-f010:**
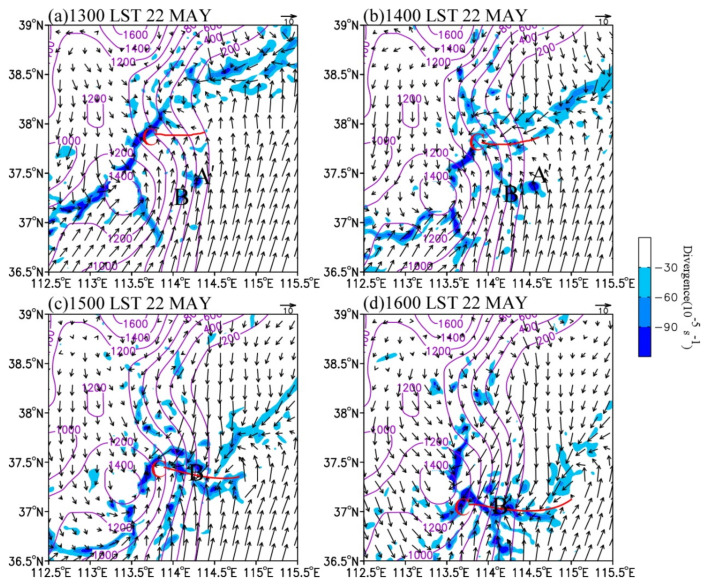
Simulated 850 hPa wind anomaly (vector, m s^−1^) and divergence (shaded, 10^−5^ s^−1^) at (**a**) 1300 LST, (**b**) 1400 LST, (**c**) 1500 LST, and (**d**) 1600 LST on 22 May 2017. The letter “C” denotes the cyclone center, and the red line indicates the position of surface convergence line.

**Figure 11 ijerph-19-09484-f011:**
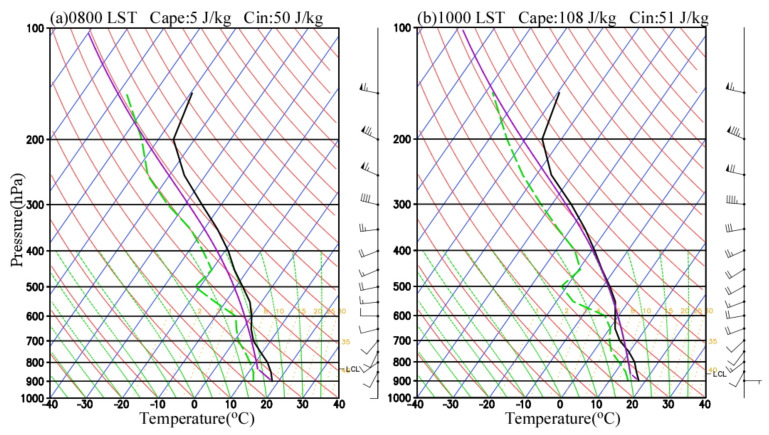
Simulated skew–T log–P diagrams at (**a**) 0800 LST and (**b**) 1000 LST on 22 May 2017 at B0877. Black and green lines denote temperature and dew-point temperature, respectively. The full (half) wind barb represents 4 m s^−1^ (2 m s^−1^).

**Figure 12 ijerph-19-09484-f012:**
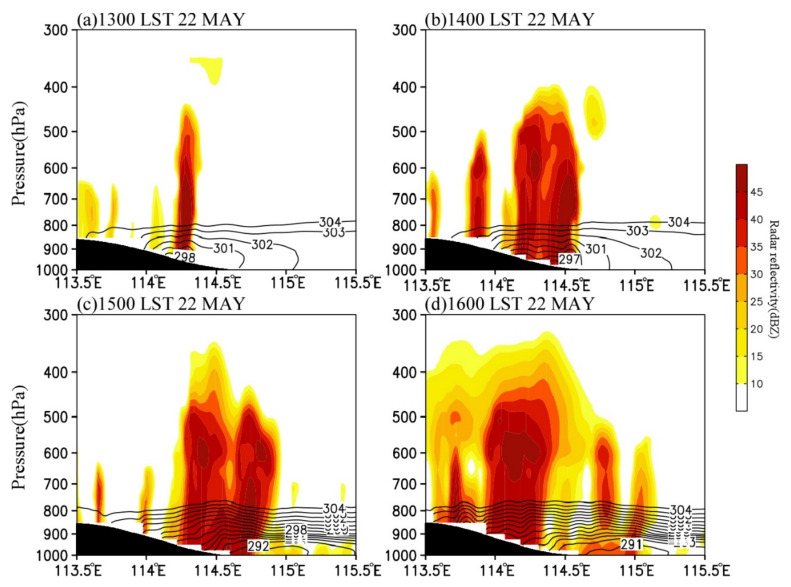
Vertical cross-sections of simulated radar reflectivity (shading, unit: dBZ) and potential temperature (contour, K) at (**a**) 1300 LST, (**b**) 1400 LST, (**c**) 1500 LST, and (**d**) 1600 LST on 22 May 2017 along the dash line in Figure 7.

**Figure 13 ijerph-19-09484-f013:**
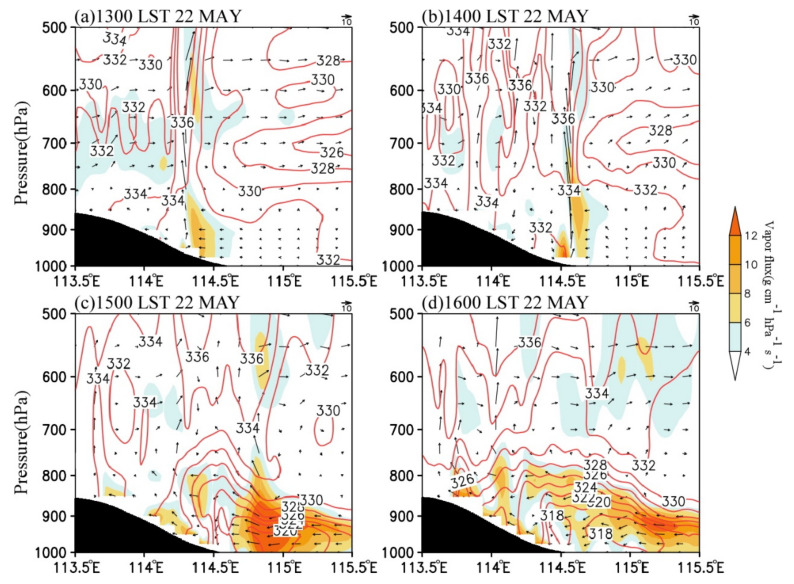
Vertical cross–sections of simulated vapor flux (shaded, g cm^−1^ hPa^−1^ s^−1^), equivalent potential temperature (contour, K), and wind field (vector, 10^−1^ m s^−1^) at (**a**) 1300 LST, (**b**) 1400 LST, (**c**) 1500 LST, and (**d**) 1600 LST on 22 May 2017. These vertical cross-sections are along the dash lines at the corresponding time, as shown in Figure 7.

**Figure 14 ijerph-19-09484-f014:**
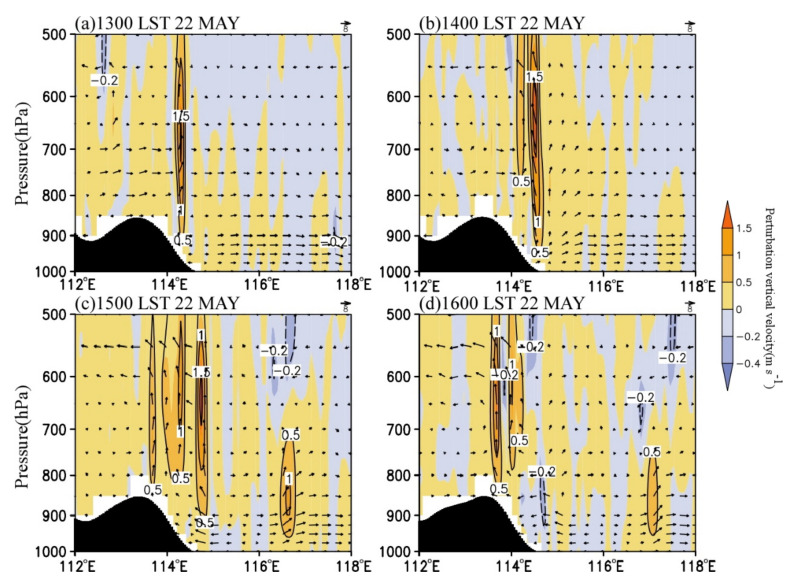
Vertical cross–sections of simulated wind anomaly (vector, 10^−1^ m s^−1^), vertical velocity (contour, unit: m s^−1^), and vertical velocity anomaly (shaded, m s^−1^) at (**a**) 1300 LST, (**b**) 1400 LST, (**c**) 1500 LST, and (**d**) 1600 LST on 22 May 2017 along the dash line in Figure 7.

**Figure 15 ijerph-19-09484-f015:**
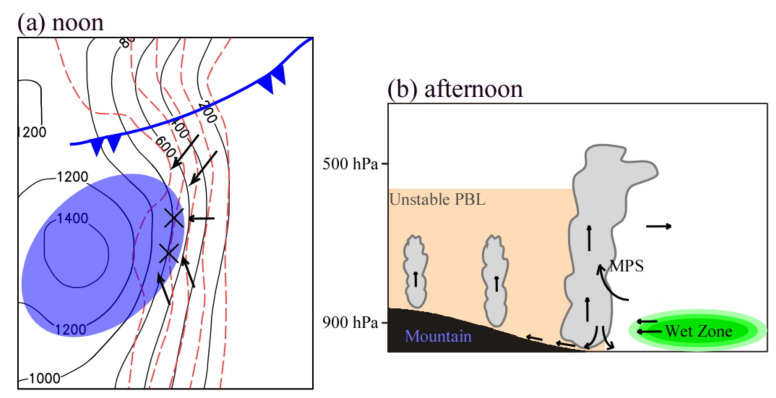
The conceptual model for the (**a**) initiation and (**b**) maintenance of convective clusters along the eastern foothills of the Taihang Mountains. Black contours represent the elevation of topography. Red dashed contours indicate near-surface temperature. The symbol “×” indicates the position of convective clusters. The blue and orange shadings denote the convergence and unstable PBL zone, respectively. The wet zone is represented by green shading. The surface cold front is shown in blue and the MPS circulation is shown with black arrows.

**Table 1 ijerph-19-09484-t001:** Dataset information.

Datasets	Spatiotemporal Resolutions	Resources
Precipitation observations	1 h	http://data.cma.cn
FNL	6 h/1° × 1°	https://rda.ucar.edu/datasets/ds083.2/index.html#sfol-wl-/data/ds083.2?g=22017
ERA5	1 h/0.25° × 0.25°	https://cds.climate.copernicus.eu/cdsapp#!/dataset/reanalysis-era5-pressure-levels?tab=form

## Data Availability

The raw data supporting the conclusions of this article will be made available by the authors, without undue reservation.

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
