# Peer review of "The Influence of a Cold Front and Topography on the Initiation and Maintenance of a Precipitation Convective System in North China: A Case Study"

_ijerph, 2022, doi:10.3390/ijerph19159484_

Round 1

Reviewer 1 Report

From my point of view, the research and the modelization developed in this paper to illustrate and analyze the rain event is very good. However, I think I'm missing a part where all that information processed separetly is presented together and combined in order to explain the main facts that lead to the occurrence of the rain event. In addition, the paper contains the concept of "topography" in the title, but throughout all the analysis topography of the mountains is not fully considered. There is a lot of information about the atmospherical aspects, which is good and interesting, but l can't see the topographic effect on the descritpion of the event. From a geographical point of view, relief results cricual in this event. Figure 1 is the only figure that shows the distribution of the relief and it is in low quality and the colors are not showing correctly the height distribution. Sbsequent figures, although intersting outputs from the modelisation, do not show the geographical charectirstics of the area of study. Figure 2 for instance has supressed the relief and the reader does not know what the black line stands for. Similarly in figure 3: it might be interesting to show the isobaric situation in China for that day but the authors could have depicted the study area and detail the isobaric map in the study area and the role of the relief. I think that although the main intention of the paper is clear, it is not fully related with the results shown. The role of the topography is limited in the accurate modelisation of the event and that should be improved.

Reviewer 2 Report

Review for “The Influence of a Cold Front and Topography on the Initiation and Maintenance of a Precipitation Convective System in North China: A Case Study”

The manuscript is generally well-structured and the topic seems to present some interesting results for readers, albeit with the standard methodology applied in this work. Overall, I suggest a "major" revision before possible consideration of the application in Int. J. Environ. Res. Public Health. My comments are listed as:

1. Although the manuscript is generally well-written, a language check by a professional native speaker or an editing agency is needed to fix some syntax, style, and phrasing problems. Only a few examples are included in my specific comments. 

2. Many grammar mistakes were found in the abstract. Please revise it carefully.

3. Please check the reference style as all references had wrong style.

4. The introduction needs to be improved and further discussion is needed. Also, please use either rainfall or precipitation in the manuscript.

5. L41-43: please rewrite

6. Many statements lack reference. Please provide them like in L49.

7. L51-55: please revise.

8. Please prepare a table to include all datasets used by the authors and their resolution and provide the source from where it was obtained (link).

9. The meaning seems to be not completed by the authors in the last sentence of page 3 (L145).

10. All figures should have legend titles with units. 

11. Figure 1 caption has information which not included in the figure. Please revise.

12. Figure 2a what are the points (B0877, ….)

13. Any figure should have one legend if they were the same as mentioned in Figure 5. 

14. Figure 5: please use only English language and include the coordinates of figures a and c.

15. No explanation should be included in the figure caption and it must be included where the figure was cited with more explanation.

I look forward to seeing a better version of the manuscript.

Reviewer 3 Report

Review on “The influence of a cold front and topography on the initiation and maintenance of a precipitation convective system in north China: A case study”

The manuscript was well organized and described. Therefore, I would like to recommend that this manuscript would be accepted in current form. 

General comments:

In this paper, The influence of a cold front and complex topography on the initiation and maintenance of the precipitation convective system occurred from May 22 to 23, 2017 were analyzed using WRF model. I think that this manuscript was well organized and explained well all contents such as datasets, model configurations, results and conclusions. One of the minor points is that authors have chosen only one rainfall event. However, authors pointed out the initiation and maintenance of the precipitation using model output, environmental conditions, and concluded that lower-level atmosphere stability of hillside and plain area, easterly anomalies were main sources of the precipitation, accordingly. The topography is one of the important sources of the rainfall system, the manuscript deals with its effect on precipitation initiation and maintenance. It would be attractive to the readers. Therefore, I would like to recommend that this manuscript would be accepted in this form.

Round 2

Reviewer 2 Report

The authors had improved the manuscript.